# Role of Pea LTPs and Abscisic Acid in Salt-Stressed Roots

**DOI:** 10.3390/biom10010015

**Published:** 2019-12-20

**Authors:** Guzel R. Akhiyarova, Ekaterina I. Finkina, Tatiana V. Ovchinnikova, Dmitry S. Veselov, Guzel R. Kudoyarova

**Affiliations:** 1Ufa Institute of Biology, Ufa Federal Research Centre, RAS, Prospekt Oktyabrya, 69, Ufa 450054, Russia; akhiyarova@rambler.ru (G.R.A.); veselov@anrb.ru (D.S.V.); 2Shemyakin–Ovchinnikov Institute of Bioorganic Chemistry, Russian Academy of Sciences, Miklukho-Maklaya Str, 16/10, Moscow 117997, Russia; finkina@mail.ru (E.I.F.); ovch@ibch.ru (T.V.O.)

**Keywords:** lipid transfer proteins, abscisic acid, salt stress, roots, pea, immunolocalization

## Abstract

Lipid transfer proteins (LTPs) are a class of small, cationic proteins that bind and transfer lipids and play an important role in plant defense. However, their precise biological role in plants under adverse conditions including salinity and possible regulation by stress hormone abscisic acid (ABA) remains unknown. In this work, we studied the localization of LTPs and ABA in the roots of pea plants using specific antibodies. Presence of LTPs was detected on the periphery of the cells mainly located in the phloem. Mild salt stress (50 mM NaCI) led to slowing plant growth and higher immunostaining for LTPs in the phloem. The deposition of suberin in Casparian bands located in the endoderma revealed with Sudan III was shown to be more intensive under salt stress and coincided with the increased LTP staining. All obtained data suggest possible functions of LTPs in pea roots. We assume that these proteins can participate in stress-induced pea root suberization or in transport of phloem lipid molecules. Salt stress increased ABA immunostaining in pea root cells but its localization was different from that of the LTPs. Thus, we failed to confirm the hypothesis regarding the direct influence of ABA on the level of LTPs in the salt-stressed root cells.

## 1. Introduction

Lipids play an important role in plant life. They enable waterproof cover that protects plants from adverse effects, perform storage and energy functions, participate in the formation of membranes, cellular differentiation, and intra- and intercellular signaling. Proteins involved in lipid transport inside the cell, apoplast, intercellular space, and phloem enable their delivery to their destination [1,2]. Lipid transfer proteins are one of the classes of plant proteins that bind and transfer lipids. A characteristic feature of the spatial structure of these proteins is the presence of a cavity covered with side radicals of hydrophobic amino acid residues [3,4]. Its presence allows the ability of plant lipid transfer proteins (LTPs) to form complexes with hydrophobic ligands including lipids and to transfer them between membranes in experiments in vitro [4,5,6]. Therefore, LTPs are believed to be involved in many processes in plants associated with lipid transport and metabolism [2]. These proteins take part in the biosynthesis of lipid polymer sporopollenin, which comprises the outer walls of spores and pollen seeds as well as in the formation of hydrophobic layers of cutin and suberin, which form protective water-impermeable barriers in different plant organs (for example, Casparian bands in roots) [7,8,9]. It has also been shown that LTPs may be involved in plant signal transduction through the systemic transport of building and signaling lipids across phloem [2]. However, direct evidence of these assumptions has not been obtained. Induction of the synthesis of LTPs in different plant organs occurs under the influence of various adverse factors on plants. Therefore, LTPs are assigned to pathogenesis related proteins (PR-proteins), which protect plants under stress [1,8,9,10]. For example, expression of *Setaria italica* SiLTP in two-week-old seedlings was induced by NaCl and polyethylene glycol, and lines over-expressing SiLTP performed better under salt and drought stresses [11]. Expression analysis in two local durum wheat varieties revealed a higher transcript accumulation of TdLTP4 in embryos and leaves under different stress conditions in the salt and drought tolerant variety compared to the sensitive one [12]. Nevertheless, it has not yet been shown in which tissues of various plant organs including roots that LTPs are localized and accumulated under stress conditions such as salinity. 

LTPs have also shown responses to stress-related plant hormones including abscisic acid (ABA) [8]. ABA is especially important for plant adaptation to abiotic stress. In particular, the ability of ABA to stimulate suberin deposition during wounding and a deficiency in mineral nutrition has been identified [13,14]. ABA has been shown to increase the expression of LTP genes. The SiLTP transcript level increased significantly after the ABA treatment in two-week-old foxtail millet seedlings [11]. ABA treatment of Arabidopsis root cultures also induced transcription of AtLtpI-4, which is involved in suberin formation in crown galls [15]. However, possible role of ABA in salt-stress induced root suberization and the participation of LTPs in this process have not been given much attention. 

In this work, we applied the immunohistochemical approach to establish tissue localization of the LTPs and the hormone ABA, both under normal conditions and salt stress, and their possible functions in pea roots.

## 2. Materials and Methods

### 2.1. Plant Growth Conditions

Seeds of the garden pea *Pisum sativum* (the cultivar “Sacharniy 2” by “Udachnye semena” company) were soaked in water for 24 h, and then wrapped with wet gauze for germination. Three-day-old pea seedlings were transplanted on rafts, placed in trays with tap water, and put on a light platform with a 14-h photoperiod, illumination of 400–500 μmol m^−2^s^−1^ PAR (ZN-500 and DNAT-400 lamps) and a temperature of 24/18 °C (day/night). At the age of four days, part of the seedlings was transferred to trays with tap water containing 50 mm sodium chloride. Solutions were changed daily.

### 2.2. Preparation of Tissue Sections

The roots were fixed for studying the immunohistochemical localization of LTPs and ABA at one and seven days after the introduction of sodium chloride into the root environment. Therefore, pieces of root tissue were taken from its central (middle part) and basal parts. Five millimeter root segments were fixed for 12 h in a solution of 4% N-(3-Dimethylaminopropyl)-N’-ethylcarbodiimide hydrochloride (Merck, Darmstadt, Germany) prepared in 0.1 M phosphate buffer (pH 7.2–7.4). Carbodiimide not only fixes proteins, but also conjugates ABA with tissue proteins [16,17]. Then, root tissues were placed in a mixture of 4% paraformaldehyde (Riedel de Haen, Seelze, Germany) and 0.1% glutaraldehyde (Sigma) for 12 h. After fixation, plant tissues were washed for 1 h in phosphate buffer and successively kept for 30 min in ethanol dilutions for their dehydration. Pieces of roots were embedded in JB4 resin (Electron Microscopy Sciences, Hatfield, PA, USA). Histological sections 1.5 μm thick were obtained using a rotary microtome (HM 325, MICROM Laborgerate, Germany).

### 2.3. Immunohistochemical Localization

Before applying the immune serum, the sections were incubated for 30 min at room temperature in phosphate buffer (50 μL per section) containing 0.2% gelatin and 0.05% Tween-20. Then, polyclonal rabbit anti-LTP (1:200 dilution) [18,19] or anti-ABA (dilution 1:80) [20] sera was applied to part of the sections. Other sections served as an immunological control, for which they were treated with non-immune rabbit serum. After washing three times in 0.1 M phosphate buffer with 0.05% Tween-20, secondary anti-rabbit IgG goat antibodies labeled with colloidal gold (Aurion, San Ramon, CA, USA) were applied (1:40 dilution). Detection of bound antibodies was carried out by applying a silver enhancer to the sections (Aurion). The appearance of the characteristic tissue blackening was observed under a light microscope. After that, silver was removed by rinsing the glasses in deionized water. The obtained preparations were shot using an AxioCam Mrc5 digital camera (Oberkochen, Carl Zeiss, Germany). Figures show typical photomicrographs representative of about nine repeats (nine stained sections of roots cut from three of either the control or NaCl-treated plants).

### 2.4. Suberin Detection

Suberin in the root tissues was stained with alcoholic solutions of Sudan III (Sigma) [21]. Suberinized tissues were stained dark orange.

## 3. Results

Pea plants were grown in conditions without or with 50 mM sodium chloride. However, even under such mild salinity, a change in plant growth rate was observed. Salt treatment of pea plants during one day did not influence plant growth, but in seven days, the treatment led to a decrease in the height of the aerial parts from 8.4 (in the control) to 6.2 cm (*n* = 10 for each variant, the difference was significant at *p* ≤ 0.05, *t*-test). One week after the start of the treatment, leaf mass was also decreased, but the values were not significantly different from the control due to characteristic variability. 

An immunohistochemical study of sections of the roots of control pea plants (without salinity) using serum containing polyclonal anti-LTP antibodies of the IgG class showed increased staining mainly around the perimeter of the cells. The observed staining developed on root sections both in the central (Figure 1A) and in the basal (Figure 1B) parts. The specificity of the method was confirmed by the absence of staining when processing sections with non-immune serum (Figure 1C,D). Along the edge of the central cylinder between the radial rows of xylem cells, the staining intensity on the LTPs was increased in the rows forming the arcs of the cells. The number of rows of such cells was greater in the transverse section of the basal part of the root, compared with the section of the central part. Immunostaining of the xylem cells for LTP was also detected.

In our experiments, one day after the addition of sodium chloride to the root medium, no differences were found in comparison with the control, when the cells of the transverse sections of the roots were stained using anti-LTP antibodies (Figure 2A should be compared with B and C with D). Only some tendency of higher staining of xylem cells was detected in the basal zone compared to the central zone of the NaCl treated plants (Figure 2B,D). The absence of changes in staining for LTP one day after the start of salt-treatment (Figure 2) may be related to the absence of morphological changes at this stage.

However, salt treatment of pea plants during the week led to a marked increase in the immune staining of cells on transverse sections of the root in its central and basal parts (Figure 3). Increased staining was noticeable in the region of the xylem, but the clearest increase was detected in the phloem.

Staining sections of pea roots with anti-ABA antibodies revealed an increase in the ABA content in plant cells during salt treatment (Figure 4). One day after the start of salt treatment, this effect was manifested in individual cells of the cortex and central cylinder, and a week later, ABA staining was higher in almost all cells of the root cross section compared to the control.

In the roots of the salt-treated plants, histochemical staining for suberin with Sudan revealed the formation of Caspari bands in the endoderma region (a clear line around the row of endodermal cells along the border between the central cylinder and the cortex) (Figure 5). This line is not noticeable on the section of the roots of the control plants.

## 4. Discussion

Three proteins of the LTP class called Ps-LTP1–3, were previously found in pea. Pea LTPs are synthesized as precursor proteins with N-terminal signal peptides and probably have extracellular localization. It was shown that genes encoding these proteins are differentially expressed in various parts of the plant. Ps-LTP1 is an abundant isoform in pea seeds and young seedlings. It was observed that pea Ps-LTP1 binds and transfers lipid molecules, forms complexes with a broad spectrum of saturated and unsaturated C12-C24 fatty acids (FA), jasmonic acid (JA), and lysolipids. This protein most effectively binds C16 and C18 unsaturated FA, which are the precursors of cutin. Therefore, Ps-LTP1 possibly takes part in lipid mobilization and biosynthesis of cutin during seed germination. Transcripts of all three Ps-LTP isoforms were found in the roots, but their biological role there is unknown [19]. 

In this paper, we investigated the tissue localization and possible functions of pea LTPs in the roots. Previous studies of tissue localization of LTPs in the roots have been performed using Arabidopsis lines expressing the glucuronidase (GUS) enzyme under the control of the LTP promoter. GUS expression under the control of the AtLtpI-4 promoter was found in the youngest cell layers of the xylem of seedlings and in the root periderm [15]. The GUS analysis showed the expression of lily [22] and foxtail millet [11] LTP genes in the root tips. In our experiments, the cells stained with anti-LTP antibodies were located predominantly along the edge of the central cylinder of the root between the radial rows of cells in the form of arcs. An increase in the number of rows of such cells was observed in the transverse section of the basal part of the root, compared with a section of its central part. This indicated that LTPs could be in the cells of a developing phloem. Detection of LTP in the periphery of the root cells (Figure 1), was in accordance with the data showing the GFP fluorescence, especially in the cytoplasm near the cell membrane of *Nicotiana benthamiana* epidermal cells transformed with the pSuper::SiLTP-GFP construct [11]. Similar experiments with wheat LPT showed that it is distributed in the peripheral cell layers and most abundant in the cell walls [12]. 

Phloem and xylem tissues together make up the transport system of higher plants. Phloem is a vascular tissue involved in the transport of products of photosynthesis, hormones, proteins, nucleic acids, metabolites, and lipids from synthesis sites to all organs of the plant [23]. Saturated and unsaturated FA, various phospholipids and hormones such as JA are present in phloem sap [24]. Recent published data indicate that such lipid-binding proteins as the homologues of Bet v 1 [24] and LTPs [25] are present there and are possibly involved in the phloem transport of hydrophobic ligands in plants. The increased staining in the phloem region revealed by us indicates the possible role of pea LTPs in the transport of lipid molecules and JA that are their ligands via the phloem or unloading of the phloem and delivery of lipids from the phloem sap to the destination. 

Xylem is responsible for the transport of water, mineral nutrients, and products from root metabolism to higher parts of the plant. However, the xylem sap also contains many proteins including metabolic enzymes, stress-related, and signal transduction proteins [26]. Such LTPs have been discovered in the xylem sap of soybean [27], tomato [28], broccoli, and rape [29]. In accordance with this, immunostaining of the root xylem suggesting the presence of pea LTPs in these cells was demonstrated by us. Possibly, as in the case of the phloem, the presence of pea LTPs in the xylem sap is due to their participation in the long-distance transport of signal molecules such as JA.

Salt stress is known to accelerate the deposition of suberin and formation of Casparian bands [30], which corresponds to the staining of suberin in the region of the root endoderma identified by us in the salt-stressed plants. It has also been shown that the induction of LTP synthesis in plants occurs under stress conditions including salt and drought stress, and this is due to the protective role of these proteins [31]. The increase in the intensity of staining in sections with anti-LTP antibodies during salt treatment in the assumed region of localization of the phloem was observed. Pea LTPs bind fatty acids of different lengths including those that are the precursors of suberin (C16–C24), and the phloem is located close to the endoderma, where the formation of Caspari bands occur. Therefore, another proposed function of pea LTPs may be to participate in root suberization under salt stress. The Casparian strips in the roots play an important role in preventing the non-selective apoplastic bypass of salts into the stele, thereby decreasing the accumulation of toxic ions under salt stress [32]. Na^+^ accumulation was lower in tobacco lines overexpressing *NtLTP4* when compared with those of wild type lines after salt treatment [33]. Our data suggest that salt-induced increase in the levels of LTP contribute to a greater deposition of suberin in the Casparian bands, bringing about increased salt tolerance.

ABA is known to be involved in the defense response of plants to abiotic stress, and, in particular, it accumulates in the tissues of salt-stressed plants [34]. The protective effect of this hormone is associated, among other things, by the induced expression of PR-protein genes [35]. In particular, ABA induces LTP synthesis in various organs of plants including roots [11,36]. At the same time, we did not find a spatial coincidence of an increased level of ABA and LTPs in the cells of individual root tissues. Earlier, a comparison of the distribution of ABA and aquaporins on a cross section of barley roots revealed their elevated levels in the same root epidermal cells and the cortical cell layer located beneath, which confirmed the participation of ABA in the regulation of the level of aquaporins [20]. No such effect was revealed in the present experiments, and, unlike LTPs, the phloem region was not distinguished by the ABA content. The tissue distribution of the hormone is most likely a consequence of the multiplicity of processes regulated to ABA and cannot directly indicate its influence on the accumulation of LTPs in pea root cells during salt treatment. The coincidence in the increased level of both ABA and LTP may have another explanation, since LTP family members have been shown to stimulate ABA biosynthesis [37].

## 5. Conclusions

In this study, we found that pea LTPs are present in plant roots and localized in phloem cells. The accumulation of LTPs in phloem tissue occurs under conditions of salt stress. This indicates the possible participation of these proteins in the formation of a water-impermeable barrier or in the transport of phloem signaling or building lipid molecules in pea roots. In order to confirm this proposal, additional studies are needed.

## Figures and Tables

**Figure 1 biomolecules-10-00015-f001:**
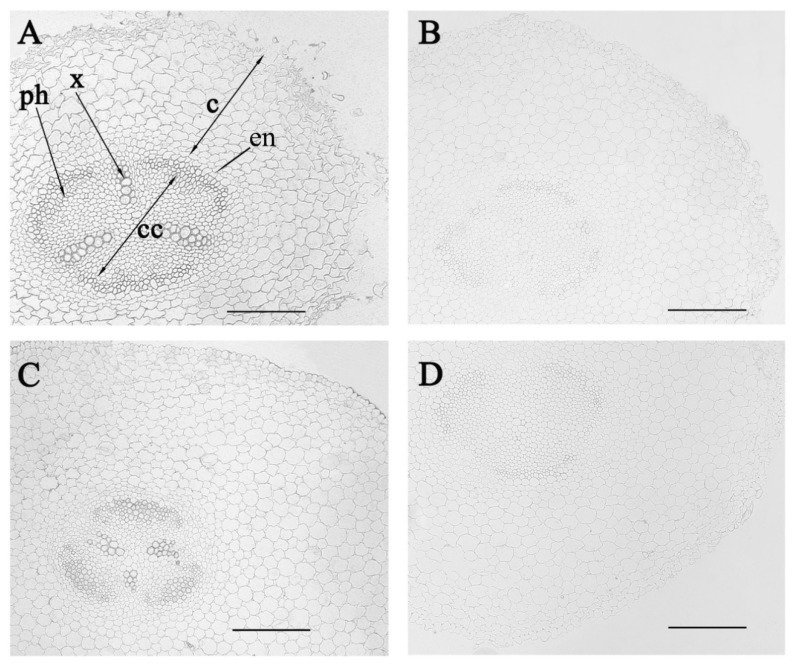
Immunolocalization of lipid transfer proteins (LTPs) in radial sections from the central (**A**,**B**) and basal (**C**,**D**) parts of 5-day-old control pea plants (not exposed to salinity) incubated with anti-LTP antibodies (**A**,**C**) and with non-immune serum (**B**,**D**). c—cortex, cc—central cylinder, x—xylem, ph—phloem, end—endoderm. Scale bar is 200 µm.

**Figure 2 biomolecules-10-00015-f002:**
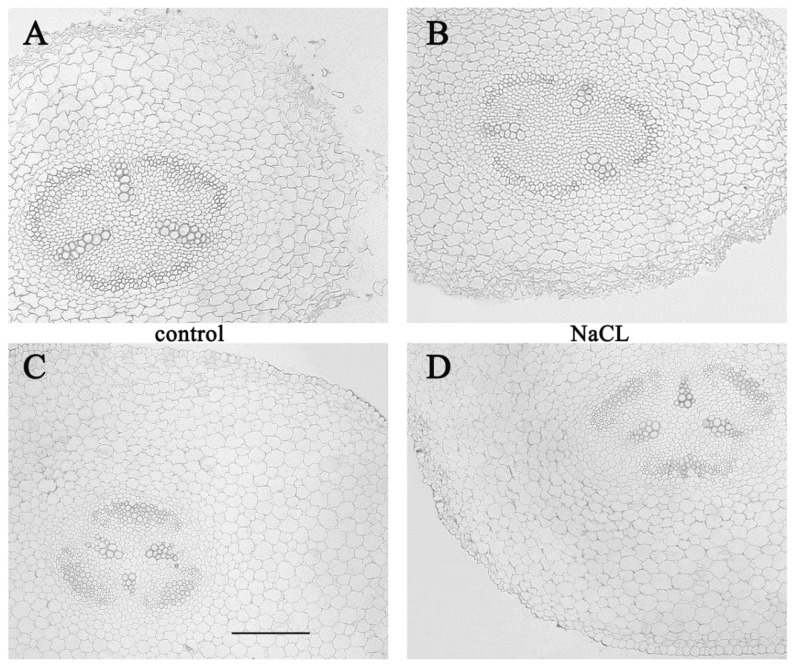
Immunolocalization of LTPs in radial sections from the central (**A**,**B**) and basal (**C**,**D**) parts of the control pea roots (**A**,**C**) and those exposed to salinity for one day (**B**,**D**). Scale bar is 200 µm.

**Figure 3 biomolecules-10-00015-f003:**
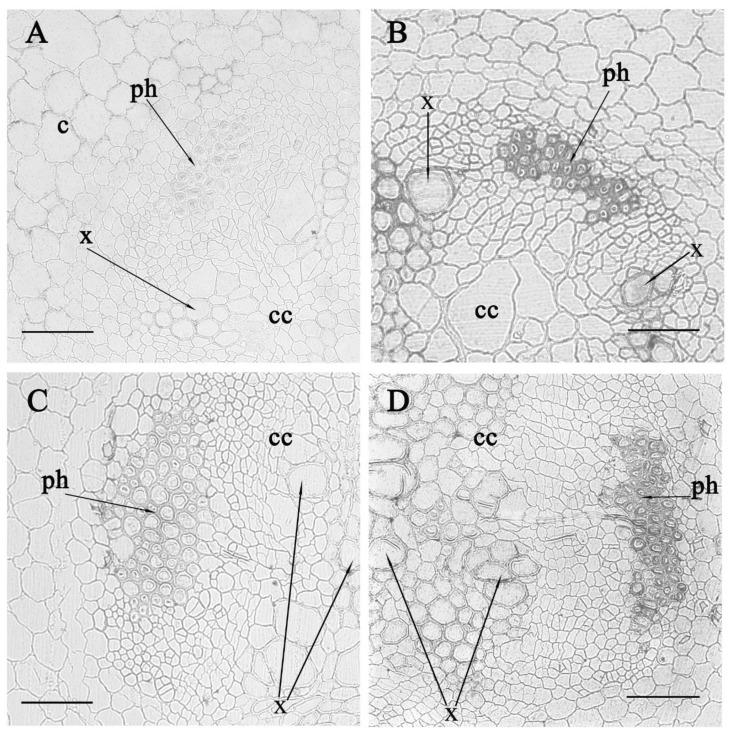
Immunolocalization of LTPs in radial sections from the central (**A**,**B**) and basal (**C**,**D**) parts of the control pea roots (**A**,**C**) and those exposed to salinity for seven days (**B**,**D**). c—cortex, cc—central cylinder, x—xylem, ph—phloem. Scale bar is 50 µm.

**Figure 4 biomolecules-10-00015-f004:**
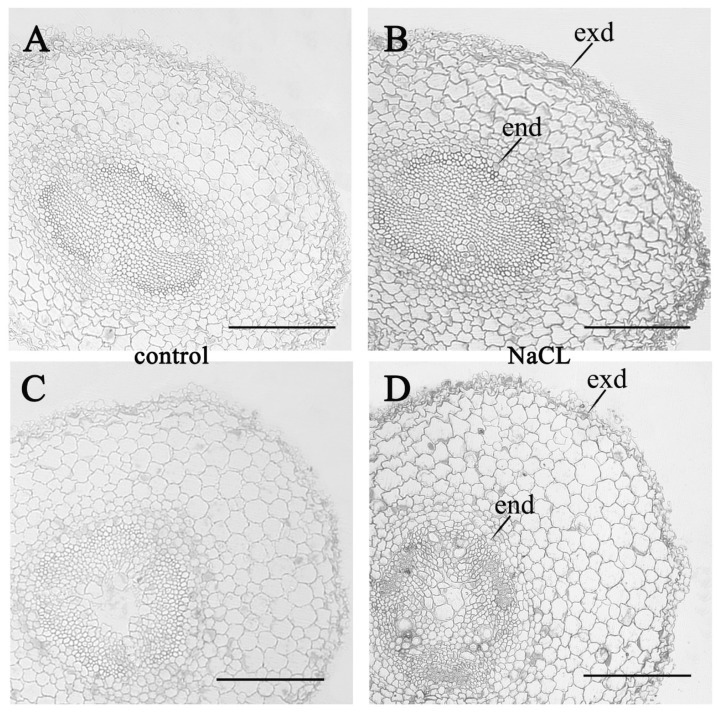
Immunolocalization of ABA with corresponding antiserum in radial sections of 5-day-old (**A**,**B**) and 11 day-old (**C**,**D**) plants grown under normal conditions (control, (**A**,**C**)) and exposed to salinity for one (**B**) and seven (**D**) days. end–endoderm, exd—exoderm. Scale bar is 200 µm.

**Figure 5 biomolecules-10-00015-f005:**
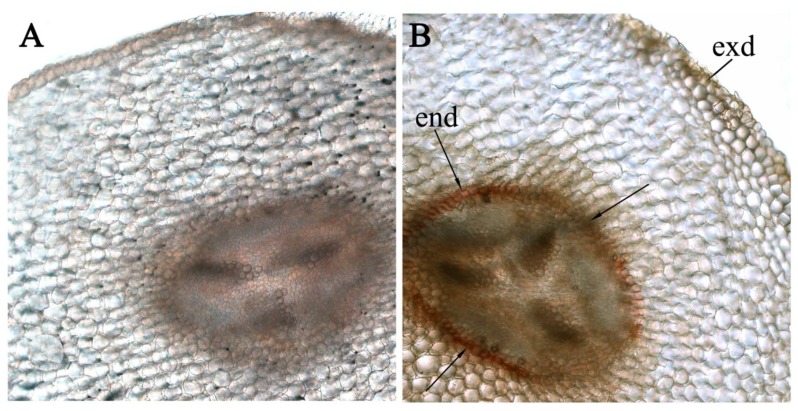
Root cross sections of pea plants grown under control conditions (**A**) and salinity (**B**) made seven days after the start of the salt-treatment and stained with Sudan III (suberin staining in the endoderma of the salt-stressed plants is indicated with arrows). end—endoderm, exd—exoderm.

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
