# Peer review of "Role of Pea LTPs and Abscisic Acid in Salt-Stressed Roots"

_biomolecules, 2019, doi:10.3390/biom10010015_

Round 1
Reviewer 1 Report
The work by Guzel R. Akhiyarova et al. investigates Lipid Transfer Protein (LTP) localization to address their possible biological function(s) in pea roots under salt stress. To identify localization of both LTPs and ABA, authors performed immunohistochemical approach using polyclonal antibodies staining revealed by colloidal gold coupled with silver enhancement. One- and seven-days NaCl (50 mM) treated pea roots were compared with control ones. Moreover, Sudan III labelling was used to correlate LTPs localization with suberin accumulation.
While the idea of the work is interesting and the attempt to consider the LTPs biological function has merit, the work suffers from some lacks. First, it is not clear where the work goes beyond previous reports that have already demonstrated the role of LTPs in improving salt stress (Pan et al., 2016, Front in Plant Science; Safi et al., 2015, Plant Physiol Biochem; …). Second, some of the analytical work is speculative, and the results should be used with more caution and/or could benefit from more appropriate figures. In that respect, I cannot recommend this manuscript for publication in its current state.
Figure 1:
The authors used an immunohistochemical study to address the LTPs localization. This set of results provide the basis for all concluding remarks. However the first figure is very challenging to interpret. The staining is considered to be significant using anti-LTP antibodies, although the comparative change between control and anti-LTP serum in the colour intensities could be seen as insufficient. Thus, the authors need to propose more contrasted figure.
I may be mistaken, but I did another interpretation of this figure. I agree with the strong staining in the (entire) stele, comparing A and C. However, I will be concerned with the colour appearing in xylem cells in B compared to D. The difference seems at least as important as the one in phloem cells. The authors should interpret this observation and/or explain the silence that accompanies this result.
The idea to confirm apoplastic localization of LTP proteins is quite interesting. However, no convincing evidence was presented in the Figure 1 to support this statement, at this scale and this resolution of observation. Thus, the authors need to propose a more defined figure. Without, the data are not robust enough to agree with the sentence “Detection of pea LTPs at the periphery of the root cells confirmed localization of these proteins in the apoplast.” (lines 160-161). Only electron microscopy can help to validate such strong assessment.
The Figure is also confusing. A and B panels do not show scale bars, are the same magnification than B or C ? Finally, the global organization (central part of the root on the left and the basal one on the right) is different from the two others figures of the manuscript (2 and 3, central part of the root on the top, basal below). Figures could be homogenized.
The authors observed i) a suberin accumulation accompanying Caspari band formation in endoderma region (Figure 5), ii) an ABA level increase (Figure 4), after salt stress. They also proposed that the latter staining intensification was particularly higher in exoderma zone (even if it is not so obvious in the figure, as I see it). These data are clumsily interpreted in the discussion part (lines 180-185), where it states “An increased level of staining of cells for ABA in the exoderma area of salt-stressed plants INDICATES that this hormone can stimulate suberin deposition and the formation of Casparian EXODERMAL bands”. A mere correlation between localization of two molecules (here ABA and suberin) is not sufficient to prove any involvement of one in the regulation of the production of the other. Moreover, and most importantly, the authors make an amalgam between endo- and exo- derma zone where suberization is both observed, but involved mechanisms differed. The authors should adjust the text to be clearer and more rigorous.
The authors analysed LTP proteins level in stele of radial sections of roots (Figure 3). Even if phloem cells labelling is clearly demonstrated in the study, out of this cannot be concluded that “The increase in the intensity of staining of sections with anti-LTP antibodies during salt treatment in the assumed region of localization of the phloem adjacent to Caspari bands was observed.” Indeed, experiments would reveal LTP accumulated in all phloem cells without any preferential accumulation (if I am wrong and it is not the case, the authors are pleased to add arrow(s) to help the reader to see what they want to focus). In any event, the array of data appears not sufficient to clearly establish such conclusion, unless additional images overweight these data.
On the subject of scientific soundness, how evaluate the significance of the results? The number of biological repetitions is not mentioned at all.
What are the effect on growth after one day of treatment? In line with these, how interpret the figure 2?
The salt-stress induced growth changes are elucidated considering height. Although this approach could allow evaluating modification rapidly and in a non-invasive assess, it has some restrictions. Since a lot of simple experiments are available to describe mechanisms involved in salt stress responses, authors have to better explain their experimental choice. Indeed, the weight, the leaf areas, etc… measurements, could permit substantial improvement on such analyses and clearly demonstrate an effect of the chosen concentration.
The text is concise. Even where they are no manuscript size recommendation, I would encourage authors to both complete and emphasize the new aspect of their work. Namely the study fails to address how the findings relate to previous research in this area, the authors should reference the related literature, especially Pan et al., 2016, Front in Plant Science; Safi et al., 2015, Plant Physiol Biochem, or more recently Gonzales et al BMC, 2017, Genomics, etc... Moreover, results will benefit to be more explained and carefully discussed.
Minor typing errors could be corrected. For example, write in italic type Latin names (line 35 and 57). Concentration of NaCl (50 mM) is sometimes considered as low (in the text)- or middle (in abstract)- salt stress.
Author Response
We are most grateful to the reviewer, who liked our work and did not find any flaws in it. And even more we are grateful to the second reviewer, who carefully supplied us with a lot valuable comments, which we tried to follow, while revising our MS.
The remarks of the second reviewer and our responses are as follows
“While the idea of the work is interesting and the attempt to consider the LTPs biological function has merit, the work suffers from some lacks. First, it is not clear where the work goes beyond previous reports that have already demonstrated the role of LTPs in improving salt stress (Pan et al., 2016, Front in Plant Science; Safi et al., 2015, Plant Physiol Biochem; …).
Our response: We are most grateful to the reviewer for supplying us with these valuable references. We got, carefully read them and used for improving Introduction and Discussion. In introduction we added short description of these publications by telling (lines 44-51): “For example, expression of Setaria italica SiLTP in two-week-old seedlings was induced by NaCl and polyethylene glycol, and lines over-expressing SiLTP performed better under salt and drought stresses [11]. Expression analysis in two local durum wheat varieties revealed a higher transcript accumulation of TdLTP4 in embryos and leaves under different stress conditions in the salt and drought tolerant variety, compared to the sensitive one [12].” Furthermore to emphasize, where our work goes beyond these reports, we added to introduction: “Nevertheless, it has not yet been shown in which tissues of various plant organs, including roots, LTPs are localized and accumulated under stress conditions such as salinity.”
“Second, some of the analytical work is speculative, and the results should be used with more caution and/or could benefit from more appropriate figures. In that respect, I cannot recommend this manuscript for publication in its current state.”
This remark is presented in greater details below and there we provide our responses
Figure 1: The authors used an immunohistochemical study to address the LTPs localization. This set of results provide the basis for all concluding remarks. However the first figure is very challenging to interpret. The staining is considered to be significant using anti-LTP antibodies, although the comparative change between control and anti-LTP serum in the colour intensities could be seen as insufficient. Thus, the authors need to propose more contrasted figure.
Our response: better control is provided, see modified fig. 1
“I may be mistaken, but I did another interpretation of this figure. I agree with the strong staining in the (entire) stele, comparing A and C. However, I will be concerned with the colour appearing in xylem cells in B compared to D. The difference seems at least as important as the one in phloem cells. The authors should interpret this observation and/or explain the silence that accompanies this result.”
Our response: Description xylem staining is introduced into Result and Discussion. We added to Result section that (line 117) “Immunostaining of xylem cells for LTP was also detected” and below that (lines 128-132) “Only some tendency of higher staining of xylem cells was detected in the basal zone compared to central zone of NaCl treated plants (Figure 2B, D).” and further: (lines 138-139): “). Increased staining was noticeable in the region of xylem, but most clear increase was detected in the phloem.” We also added to the Discussion section that (lines 200-206) “Xylem is responsible for transport of water, mineral nutrients, and products from root metabolism to higher parts of the plant. But, the xylem sap also contains many proteins, including metabolic enzymes, stress-related and signal transduction proteins [26]. Such LTPs were discovered in xylem sap of soybean [27], tomato [28], broccoli and rape [29]. In accordance with this immunostaining of root xylem suggesting the presence of pea LTPs in these cells, was demonstrated by us. Possibly, as in the case of the phloem, the presence of pea LTPs in the xylem sap is due to their participation in long-distance transport of such signal molecules as JA.”
“The idea to confirm apoplastic localization of LTP proteins is quite interesting. However, no convincing evidence was presented in the Figure 1 to support this statement, at this scale and this resolution of observation. Thus, the authors need to propose a more defined figure. Without, the data are not robust enough to agree with the sentence “Detection of pea LTPs at the periphery of the root cells confirmed localization of these proteins in the apoplast.” (lines 160-161). Only electron microscopy can help to validate such strong assessment.”
Our response: We accept criticism of reviewer and introduced corresponding changes. We deleted that “Detection of pea LTPs at the periphery of the root cells confirmed localization of these proteins in the apoplast“. Instead we just related this result to those obtained by Pan et al. and Safi et al, recommended by reviewer. This fragment in its present form sounds as follows (lines 186-190): “Detection of LTP in the periphery of root cells (Figure 1), is in accordance with the data showing the GFP fluorescence especially in the cytoplasm near the cell membrane of Nicotiana benthamiana epidermal cells transformed with pSuper::SiLTP-GFP construct [11]. Similar experiments with wheat LPT showed that it is distributed in the peripheral cell layers and most abundant in the cell walls [12].
“The Figure is also confusing. A and B panels do not show scale bars, are the same magnification than B or C ? Finally, the global organization (central part of the root on the left and the basal one on the right) is different from the two others figures of the manuscript (2 and 3, central part of the root on the top, basal below). Figures could be homogenized”
Our response: Figures were homogenized
“The authors observed i) a suberin accumulation accompanying Caspari band formation in endoderma region (Figure 5), ii) an ABA level increase (Figure 4), after salt stress. They also proposed that the latter staining intensification was particularly higher in exoderma zone (even if it is not so obvious in the figure, as I see it). These data are clumsily interpreted in the discussion part (lines 180-185), where it states “An increased level of staining of cells for ABA in the exoderma area of salt-stressed plants INDICATES that this hormone can stimulate suberin deposition and the formation of Casparian EXODERMAL bands”. A mere correlation between localization of two molecules (here ABA and suberin) is not sufficient to prove any involvement of one in the regulation of the production of the other. Moreover, and most importantly, the authors make an amalgam between endo- and exo- derma zone where suberization is both observed, but involved mechanisms differed. The authors should adjust the text to be clearer and more rigorous”
Our response: we are sorry for “amalgam”. We did mixed these terms in conclusion. This was rectified. We deleted the sentence “An increased level of staining of cells for ABA in the exoderma area of salt-stressed plants indicates that this hormone can stimulate suberin deposition and the formation of Casparian exodermal bands” and introduced indication of exoderma and endoperma on the figures.
“The authors analysed LTP proteins level in stele of radial sections of roots (Figure 3). Even if phloem cells labelling is clearly demonstrated in the study, out of this cannot be concluded that “The increase in the intensity of staining of sections with anti-LTP antibodies during salt treatment in the assumed region of localization of the phloem adjacent to Caspari bands was observed.” Indeed, experiments would reveal LTP accumulated in all phloem cells without any preferential accumulation (if I am wrong and it is not the case, the authors are pleased to add arrow(s) to help the reader to see what they want to focus). In any event, the array of data appears not sufficient to clearly establish such conclusion, unless additional images overweight these data. “
Our response: We were not very clear in what we wanted to say (sorry, but English is not our first language). We tried to be more exact and rephrased the paragraph on this theme adding some more references. It now sounds differently and hopefully clearer (lines 211-221): “The increase in the intensity of staining of sections with anti-LTP antibodies during salt treatment in the assumed region of localization of the phloem was observed. Рea LTPs bind fatty acids of different length, including those that are the precursors of suberin (C16-C24), and the phloem is located close to endoderma, where formation of Caspari bands occur. Therefore, another proposed function of pea LTPs may be to participate in root suberization under salt stress. The Casparian strips in the roots play an important role in preventing the non-selective apoplastic bypass of salts into the stele thereby decreasing accumulation of toxic ions under salt stress [32]. Na+ accumulation was lower in tobacco lines overexpressing NtLTP4 compared with those of WT lines after salt treatment [33]. Our data suggest that salt-induced increase in the levels of LTP contribute to greater deposition of suberin in Casparian bands bringing about increased salt tolerance.”
“On the subject of scientific soundness, how evaluate the significance of the results? The number of biological repetitions is not mentioned at all.”
Our response: In accordance with this remark we added that (lines 96-97) “Figures show typical photomicrographs representative of about 9 repeats (9 stained sections of roots cut from 3 either control or NaCl-treated plants)”
“What are the effect on growth after one day of treatment? In line with these, how interpret the figure 2?”
Our response: In accordance with this remark we added that (line 104)“. Salt treatment of pea plants during one day did not influence plant growth” and below (lines 130-132)– “The absence of changes in staining for LTP one day after the start of salt-treatment (Figure 2) may be related to the absence morphological changes at this stage.”
“The salt-stress induced growth changes are elucidated considering height. Although this approach could allow evaluating modification rapidly and in a non-invasive assess, it has some restrictions. Since a lot of simple experiments are available to describe mechanisms involved in salt stress responses, authors have to better explain their experimental choice. Indeed, the weight, the leaf areas, etc… measurements, could permit substantial improvement on such analyses and clearly demonstrate an effect of the chosen concentration.”
Our response: In accordance with this remark we added that (lines 106-108) “One week after the start of the treatment, leaf mass was also decreased, but the values were not significantly different from the control due to characteristics variability.”
“The text is concise. Even where they are no manuscript size recommendation, I would encourage authors to both complete and emphasize the new aspect of their work. Namely the study fails to address how the findings relate to previous research in this area, the authors should reference the related literature, especially Pan et al., 2016, Front in Plant Science; Safi et al., 2015, Plant Physiol Biochem, or more recently Gonzales et al BMC, 2017, Genomics, etc... Moreover, results will benefit to be more explained and carefully discussed.”
Our response: Thanks for suggestions. As we mentioned above, we found the publications of Pan et al. [11] and Safi et al [12] and used them for improving Discussion. Examples are listed above. We also found and cited Gonzales (lines 236-237): “The coincidence in the increased level of both ABA and LTP may have another explanation, since LTP family members have been shown to stimulate ABA biosynthesis [37].” Furthermore we included into discussion the reports of Buhtz et al. [29], Chen et al. [32] and Xu et al. [33] (see above).
“Minor typing errors could be corrected. For example, write in italic type Latin names (line 35 and 57). Concentration of NaCl (50 mM) is sometimes considered as low (in the text)- or middle (in abstract)- salt stress.”
Our response: We corrected Latin names in Italic type and described 50 mM NaCl as mild stress throughout the text
Reviewer 2 Report
The manuscript details an immunohistological approach to investigate the tissue localisation of LTPs and the plant hormone ABA in the roots of pea plants exposed to salt stress, relative to a control. The results of these experiments are correlated with the observed distribution of suberin, as revealed by Sudan III staining.
The manuscript appears concise and well presented, providing visual data to describe the observed experimental results. The manuscript adds new information to address the postulated linkage between the direct influence of ABA on LTPs in plant roots during stress responses of pea plants.
Author Response
We are most grateful to the reviewer, who liked our work and did not find any flaws in it.
Round 2
Reviewer 1 Report
I would thank authors taking into account my remarks. I just have now a minor typing error; in figure 1, “en” should be “end”.